# Shaped-Based Tightly Coupled IMU/Camera Object-Level SLAM

**DOI:** 10.3390/s23187958

**Published:** 2023-09-18

**Authors:** Ilyar Asl Sabbaghian Hokmabadi, Mengchi Ai, Naser El-Sheimy

**Affiliations:** Department of Geomatics Engineering, University of Calgary, Calgary, AB T2N 1N4, Canada; mengchi.ai@ucalgary.ca (M.A.); elsheimy@ucalgary.ca (N.E.-S.)

**Keywords:** object-level SLAM, RBPF-SLAM, shape-based pose estimation, undelayed initialization, IMU/camera fusion, tightly coupled, coarse-to-fine pose estimation

## Abstract

Object-level simultaneous localization and mapping (SLAM) has gained popularity in recent years since it can provide a means for intelligent robot-to-environment interactions. However, most of these methods assume that the distribution of the errors is Gaussian. This assumption is not valid under many circumstances. Further, these methods use a delayed initialization of the objects in the map. During this delayed period, the solution relies on the motion model provided by an inertial measurement unit (IMU). Unfortunately, the errors tend to accumulate quickly due to the dead-reckoning nature of these motion models. Finally, the current solutions depend on a set of salient features on the object’s surface and not the object’s shape. This research proposes an accurate object-level solution to the SLAM problem with a 4.1 to 13.1 cm error in the position (0.005 to 0.021 of the total path). The developed solution is based on Rao–Blackwellized Particle Filtering (RBPF) that does not assume any predefined error distribution for the parameters. Further, the solution relies on the shape and thus can be used for objects that lack texture on their surface. Finally, the developed tightly coupled IMU/camera solution is based on an undelayed initialization of the objects in the map.

## 1. Introduction

The classical solutions to the SLAM problem rely on geometrical primitives (such as points, lines and planes) [1,2,3]. However, one of the challenges with these approaches is that such simple forms cannot be used for intelligent interactions of a robot with the environment. More recently, object-level methods have emerged as an alternative solution to the SLAM problem. In contrast to the classical methods, these solutions seek to represent the map of an environment using semantic objects. However, unlike simpler geometrical forms, objects can be difficult to represent and require novel techniques to incorporate them into the solution to the SLAM problem. Some of the important aspects and research gaps are reviewed in the following passages.

A monocular camera is one of the important sensors for object detection. This sensor provides abundant information (e.g., colour) and is often included in many platforms, such as mobile wheeled robots and handheld devices. A monocular camera suffers from scale ambiguity. Due to this ambiguity, this type of camera cannot observe the distance from the object of interest and, more importantly, cannot estimate the trajectory with a real scale. In order to mitigate this issue, the fusion of a monocular camera with integrated IMU/Global Navigation Satellite Systems (GNSS) platforms [4,5] can be considered. In these platforms, the GNSS signals provide solution updates to the IMU, which helps avoid the accumulation of errors in the mechanization process. Unfortunately, GNSS signals are not available in most indoor environments. Therefore, the updates should be provided using cameras. Such updates are available if an initialized object (in the map) is observed by the camera. This paper proposes an IMU/camera fusion technique using undelayed object initialization. This initialization can provide immediate updates to the device’s position after the object is observed only in one image.

IMU and monocular camera fusion can be achieved using tightly and loosely coupled methods. The main distinction between the two approaches is how solution updates are performed. While in a loosely coupled method, an independent pose estimation using a camera is required for the update, in a tightly coupled method, updates are provided by directly evaluating observation likelihood [6]. In the past, both loosely [7] and tightly coupled [8,9] methods for IMU/camera fusion were proposed. However, tightly coupled fusion is more advantageous for an object-level solution. One reason for this advantage is that a loosely coupled approach depends on independent camera pose estimation using a single image. The error in this pose will ultimately lead to inaccuracies in the solution, specifically if the object of interest is severely occluded in some images. In order to address this issue, this research proposes a novel, object-level, tightly coupled fusion.

Object representation refers to prior information about the shape and appearance of an object. Object representation alongside detection algorithms are two key aspects of object-level solutions. Many object-level solutions rely on the detection of salient feature points on an object [10,11,12,13]. However, most objects lack such feature points. Further, the detection of these salient features under different illumination conditions can be difficult. In this research, we developed a solution to the SLAM that only relies on an object’s shape. Object-level solutions based on shape have recently been proposed [14]. However, these solutions are not based on IMU/camera fusion.

The estimation of the uncertainties of the parameters is a key aspect of most state-of-the-art solutions to the SLAM problem. These uncertainties are assumed to have Gaussian distributions [15,16,17,18]. However, this assumption is not realistic in many circumstances. For example, symmetrical objects can produce similar silhouettes in the images from different viewpoints, which will result in pose ambiguities. These ambiguities cannot be accurately represented using Gaussian distribution. In this research, a solution based on RBPF is developed that does not rely on such assumptions for the uncertainties. RBPF-SLAM has been extended to the objects [19]; however, such extension is not based on an object’s shape.

The main objective of this research is to investigate the possibility of developing an accurate solution to the SLAM relying on an object’s shape. For this solution, it is assumed that no salient feature points are available or can be detected on the object’s surface. The proposed method is based on novel techniques of object initialization and particle weight updates that rely on the contour of the segmented object in the image. In this study, the influence of background clutter, the trajectory length, and the presence of occlusions on the accuracy (and other assessment metrics) are investigated. This method is tested using a handheld device. The contribution of this research can be summarized as follows:A shape-based solution to the SLAM problem that can achieve a 4.1 to 13.1 cm error (in an indoor environment) is developed.An object-level tightly coupled IMU/camera fusion is developed. The particle weight update does not require point-to-point data correspondences and relies only on the contour of the segmented object.An undelayed object initialization is developed. This method mitigates error accumulation due to IMU mechanization. The undelayed initialization is achieved using a novel coarse-to-fine pose estimation.

Section 2.1 and Section 2.2 review the literature regarding object-level mapping/localization frameworks and object representation methods. The methodology of the developed solution is provided in Section 3. The results and discussions are provided in Section 4. Finally, the conclusions and possible future improvements are provided in Section 5.

## 2. Literature Review

### 2.1. Object-Level Mapping and Localization Frameworks

The earliest solutions to the object-level SLAM were based on extended Kalman filtering (EKF) [20,21,22]. One challenge with EKF is that as a dynamic Bayesian network (DBN), this method processes the observations one at a time. Therefore, the propagation of error can lead to a reduction in the accuracy of the estimation. In order to address this challenge, more recent object-level solutions relied on the frontend/backend paradigm. In the frontend, an initial estimate of the solution is obtained using DBN or other techniques (one possible alternative to the DBN is to use keyframe-based local bundle adjustment (LBA) [23]). The initial solution is further optimized in the backend using global bundle adjustment (GBA), factor graphs (FG) [24], and others.

Current frameworks for object-level solutions can be categorized into two groups. The first group uses the obtained semantic information of the objects of interest to increase the accuracy of the classical solutions. A possible improvement can be achieved by using semantic information to improve the data correspondence. For example, in classical solutions to the SLAM problem, features such as FAST and Rotated BRIEF (ORB) [25] and scale-invariant feature transformation (SIFT) [26] are often used for data correspondence. In some scenarios, false correspondence can occur if the two features have an identical visual signature but do not correspond to the same landmark on the map. However, in most cases, these features are detected on different objects. Therefore, adding a processing step of object class detection can result in avoiding false correspondences. Methods in this first group of object-level solutions often use LBA in the frontend and FG (or GBA) [10,12,16,27] in the backend. These approaches only use semantic information to improve the classical methods and, therefore, can be considered an extension of the classical solutions to the SLAM problem.

In the second group of solutions, the mapping and localization are performed by relying on the objects only. In these methods, the objects are inserted into the map with six degrees of freedom (6DoF). In this category, methods have represented the objects in the map using ellipsoids [28,29,30], cuboids [31], and other forms. The geometrical forms of representation, such as ellipsoids, have the advantage that they can provide a simple observation model (defined as the mathematical model of the observations and the parameters to be estimated). Such models can be easily integrated into the process of FG [30] and GBA [29,31]. However, most of the current methods in the second group rely on an initially estimated position using classical solutions such as ORB SLAM 2 [32].

Both groups of solutions assume that the errors are Gaussian or can be approximated as a Gaussian distribution. However, this assumption is not suitable in many circumstances. In order to incorporate other types of error, RBPF-SLAM is introduced in the literature. RBPF-SLAM was originally implemented using rangefinders [33]. Later, it was also implemented using cameras. These implementations use points [34], lines [35], and, more recently, objects [19]. However, the method in [19] only relies on a monocular camera. The developed solution in this paper utilizes an IMU to predict the position of the particles in the next frame, while the images obtained using the camera are used to update the weight of the particles. The sensor fusion is achieved in a tightly coupled fashion where the particle weighing is directly performed using the observation likelihood.

Unfortunately, the fusion of an IMU and a monocular cannot resolve the scale ambiguity using only a single observation. Thus, the landmarks (e.g., points, lines, objects) observed for the first time in the images cannot be inserted into the map with low uncertainty. In order to address this issue, delayed initialization has developed in the past [36,37]. In the delay initialization, the landmarks should be observed from different viewpoints. During the delayed period, the solution should remain dependent on the other sensors (e.g., IMU) if such sensors are available. If no other sensor is available, the solution should rely on a motion model heuristic (e.g., constant velocity [1]). Both types of solutions will result in the accumulation of errors typical of dead reckoning systems.

In order to address this issue, undelayed initialization methods have also been proposed in the past [38,39]. In an undelayed initialization, the landmark is inserted into the map with large uncertainty using the first image that is observed. It is reported that undelayed initialization can be used to provide a partial update to the estimated position and thus improve the accuracy of the trajectory estimation [34]. The two approaches are compared in Figure 1. In this figure, for the delayed initialization, the landmark cannot be inserted into the robot’s map using one observation at a single epoch ($t_{k}$). Thus, the initialization should be delayed until the object is also observed from a different viewpoint (such as $t_{k+1}$). During this period, no updates are available to the robot’s trajectory using the observations. In contrast, in the undelayed initialization, the landmark is inserted into the map in the first frame in which they are observed ($t_{k}$). As the robot moves and observes the object of interest, the uncertainty in the location of the landmark can decrease. The challenge with undelayed initialization is that the uncertainty of the pose of the landmark is very large initially, and thus, a large number of particles are required to be sampled to ensure high accuracy of the estimation. In this research, a novel undelayed initialization is developed. This initialization of an object uses deep learning-based pose estimation to reduce the uncertainty in the device’s pose to an unknown distance of the camera from the objects. Further objects, unlike geometrical primitives (such as points, lines, and planes), can be assumed to have approximately known dimensions. Such an assumption helps reduce the scale ambiguity. This is explained more in Section 3.4.

The abovementioned methods rely on other important components in the object-level SLAM. One of these components is object representation. In this research work, the objects are represented using their shapes. The advantage of this approach over the state of the art is explained in the following section.

### 2.2. Object Representation

Object representation refers to prior information about the appearance and/or the shape of an object. The representation is required in many other components of an object-level SLAM. For example, object pose estimation is often performed by matching the detected/segmented object to this representation. The classical solutions and the state of the solutions often rely on representing objects using salient feature points [10,12,22]. These feature points require texture on the surface of an object. Unfortunately, in real-life scenarios, many objects lack texture. Another ubiquitous object representation is based on simpler geometrical shapes such as ellipsoids [29], cuboids [31], and others. These geometrical primitives provide the advantage of a simple observation model. However, such shapes cannot represent most objects accurately. In contrast to these simple shapes, objects can also be represented in detailed 3D shape models. The solutions that utilize such models can be divided into online [40,41,42] and offline [43,44,45,46] methods. In the online methods, the 3D model of an object is jointing estimated with the solution to the SLAM. This approach is computationally very costly. In contrast, in the offline methods, an object model is built prior to the experiments. However, matching 3D shape priors to the 2D segmented contour in the image can be challenging (and such models are more suitable for RGB-D cameras). One possible solution is using an initial estimated trajectory of the camera to build a point cloud. The 3D shape priors can be used to detect and estimate the poses of the objects captured in such point clouds [47]. The second solution is to represent an object as a 2D shape priors set, which is more suitable for a monocular camera. Such sets are built by capturing an object from different viewpoints [48]. Unfortunately, object representation using images lacks robustness to the in-plane translation, rotation, and scale variations. Such geometrical variations are anticipated to be encountered when matching the segmented object in the image to the object representation. In order to increase the robustness of the method to geometrical variations, the 2D images of the objects are transformed into parameterized 2D contours [49]. It is shown that these shape sets are capable of providing coarse pose estimation for objects in the camera frame. This research uses the method in [49] for object representation. Representing an object using its shape prior requires novel particle weighting and landmark initialization approaches. This will be explained in detail in Section 3.2 and Section 3.3.

A summary of some object-level solutions is provided in Table 1. These methods are categorized based on the type of mapping/localization framework, object representation model, and object detection/segmentation method. The object detection/segmentation methods in the literature are often based on deep neural networks (DNN) (e.g., YOLO [50]) or classical feature detectors/descriptors (e.g., SIFT). Some solutions in the literature rely on decoupled frontend estimation, where first the camera trajectory and second the object poses are estimated. Most current object-level solutions rely on ORB-SLAM 2 for an initial trajectory and map estimation. This dependency can cause several restrictions (for example, due to the reliance of ORB-SLAM 2 on salient feature point detection). In contrast to these solutions, the developed method does not rely on ORB-SLAM 2.

## 3. Methodology

### 3.1. Overview

In the past, DBN has been suggested as a possible solution to the SLAM problem. An overview of DBN is shown in Figure 2. In this figure, the landmarks, the robot’s states, the inputs to the robot, and the observations are denoted as m, x, u, and z. In Figure 2, the known variables are shown in white circles. These include the observations (such as images) or the input (such as the motion commands sent to a robot). Further, in Figure 2, the hidden variables are shown in gray circles. The hidden variables include the robot’s trajectory and the landmarks’ positions. These variables are not directly observed but can be inferred using the known variable. The goal in formalizing a solution to the SLAM problem using DBN is to estimate these hidden variables. In this research, each landmark is represented by six parameters. Three parameters are used to represent the position of a landmark, and three parameters are used to represent the orientation of the landmark. In contrast, in the classical point-based solutions to the SLAM, the landmarks are represented only using three parameters for the position.

Every DBN has two steps: prediction and update. In the prediction, a motion model is used to estimate the pose of a robot or a device. This motion model can be obtained using IMU mechanization [51], kinematic modelling of a robot, or motion heuristics (such as constant velocity or constant acceleration models). In the update step, the estimated pose of the robot is corrected with the help of an observation model. The observations can be images obtained from sensors such as monocular cameras, point clouds obtained using light detection and ranging (LiDAR), and others. The two steps of prediction and update are repeated as more observations become available to the robot. DBN can be solved using many approaches. These include solutions based on EKF and particle filter (PF). PF-based solutions can be applied in a wide variety of circumstances where the overall distribution of the errors is not known beforehand. In contrast, methods such as EKF assume that the errors have a Gaussian distribution. However, the computational cost of implementing EKF is lower than PF.

PF-based solutions suffer from the curse of dimensionality (COD) [52]. The exact definition of COD is not within the scope of this research, and relevant research [53] can be investigated for more information. Due to the COD, increasing the number of dimensions results in an increase in the number of particles required to sample the solution space. Since in the SLAM problem, the number of unknowns (the robot’s trajectory and the landmarks’ poses) is often large, the number of particles should be increased substantially. Therefore, in order to implement PF, it is important to address the COD problem. One possible solution is to use RBPF, which is a computationally more efficient implementation of PF. RBPF has been used in the past as a solution to the SLAM problem. In RBPF-SLAM, similarly to PF-SLAM, the state of the robot and the state of the landmarks are represented using particles. However, there are key differences between the two approaches, and RBPF-SLAM takes advantage of the structure of the SLAM. Due to this structure, for a given particle, the errors in the pose of the landmarks can be considered conditionally independent [33]. Therefore, the uncertainties in one landmark can be processed and stored independently of the other landmarks for a given particle. Such an approach can lead to a reduction in the required amount of computations.

In order to explain RBPF-SLAM mathematically, the notation in [54] is followed. Equation (1) shows the DBN for the SLAM problem formalized using RBPF-SLAM [54]. In this equation, the symbol $x_{1:t}$ corresponds to the estimated trajectory of the robot up to time step t. The a corresponds to the data association, and other symbols are consistent with their definitions provided earlier in this section. Data association is an important problem in the solutions to SLAM. Data association refers to the task of assigning observations to landmarks. An incorrect assignment will result in large errors and a possibility of the failure of the solution. In order to understand Equation (1), the right-hand side can be considered as two terms. The first term represents the posterior of the robot poses, and the second term represents the posterior of the map. It can be seen that for a given trajectory ($x_{1:t}$), the posterior of each landmark is considered to be independent in RBPF-SLAM.
(1)$$p\left(x_{1:t},m|a_{1:t},z_{1:t},u_{1:t}\right)=\;p\left(x_{1:t}|a_{1:t},z_{1:t},u_{1:t}\right)\prod_{j}p(m_{j}|x_{1:t},a_{1:t},z_{1:t},u_{1:t})$$

In RBPF-SLAM, as implemented in [54], the data association is determined using a maximum likelihood estimation (MLE) shown in Equation (2) (the superscript [n] corresponds to the weight of the $n^{th}$ particle). The ${\hat{a}}_{t}^{\left[n\right]}$ can be inserted in Equation (1) once it is known. In this research, the possibility of false data association is reduced by avoiding one-to-one point data correspondence. This is explained in detail in Section 3.3.
(2)$${\hat{a}}_{t}^{\left[n\right]}=\;{argmax}_{a_{t}}p(z_{t}|a_{t},{\hat{a}}_{1:t-1}^{\left[n\right]},x_{1:t}^{\left[n\right]},z_{1:t-1},u_{1:t})$$

In order to update the estimated trajectory and the map, the particles are weighted (and resampled) in an RBPF-based solution. The particle weighting can be achieved using observation likelihood. This can be seen in Equation (3).
(3)$$w_{t}^{\left[n\right]}\propto p\left(z_{t}|{\hat{a}}_{t}^{\left[n\right]},x_{1:t}^{\left[n\right]},z_{1:t-1},u_{1:t}\right)$$

If the weight update (and, in general, the update step in any DBN) is achieved directly using the observation likelihood (Equation (3)), then the developed approach can be considered a tightly coupled fusion of IMU and monocular camera [6]. If the weight update is obtained after an independent estimation of the robot’s pose using the camera and the IMU, then the method is deemed a loosely coupled approach [6]. Since Equation (3) is directly evaluated in this research, the developed method can be considered a tightly coupled solution. By way of summary, the following four modules should be implemented:The proposal distribution corresponds to the predicted state of the robot in a SLAM problem. The proposal distribution can be obtained using the motion model of a robot or a device. In this research, such a motion model is provided using IMU mechanization.Particle weighting corresponds to the update step in DBN. The particles are weighted using observation likelihood. In this research, the actual and the predicted observations are obtained using semantic segmentation of the object and the predicted projection (onto the camera) of the object, respectively. This is explained in more detail in Section 3.2 and Section 3.3.Particle resampling is an important step in any PF-based solution. In the resampling, the particles with higher weights are duplicated, while particles with lower weights are discarded. In this research, classical sequential importance resampling (SIR) [55] is used.Based on the abovementioned processes, landmark initialization is another important topic that should be addressed in every solution to the SLAM problem, which is explained in Section 3.4.

### 3.2. Tightly Coupled IMU/Camera Fusion

In the following, an overview of the tightly coupled fusion of IMU and a monocular camera is provided. The details about the mathematical derivation are also presented later in this section. The flowchart of the developed solution can be seen in Figure 3. The algorithm starts by initializing the particles in the map. Each particle includes an estimated pose of the device and poses of the objects (the particle initialization process is very similar to the landmark initialization, and it will be explained more in Section 3.4). It is assumed that at least one object is visible in the first image in order to achieve particle initialization. The segmented object in this image is used to estimate the pose of the camera. Once the camera pose is estimated up to an unknown scale, the particles are sampled around this pose with predetermined uncertainties in each direction. The uncertainties in this step should be provided by the user. It is important to note that the predetermined uncertainty should be larger in the direction from the camera’s center to the object (due to the explained scale ambiguity of the monocular camera). The other directions can be assigned with lower uncertainty. In the experiments, we have assigned 20 cm (standard deviation) in the direction from the camera center to the object and 5 cm (standard deviation) in other directions. However, uncertainties depend on the overall sizes of the objects used in the process of SLAM.

Once the particles are initialized in the map, their predicted positions are estimated using the IMU mechanization, as shown in Figure 3. This research follows the steps in [48] for mechanization, and it is assumed that the coordinate frame of the device (body frame) coincides with the IMU’s frame. The initial position of the IMU is denoted as the inertial frame, and the robot’s trajectory is estimated in this frame. The mechanization will provide a predicted pose of the device in the inertial frame. Since mechanization is deterministic, a process noise term should be added to the predicted pose of the particles. Process noise is a very important consideration in incorporating uncertainties in the motion model. Such uncertainties can be due to errors in the readings of the IMU (both accelerometers and the gyroscopes). In order to include the process noise, the estimated noise variance for each of the sensors can be utilized. In this research, the noise term is sampled from a distribution with a variance corresponding to the provided information by the manufacturer of the IMU. This sampled value is added as noise to the raw measurement. The output of this step is the proposal distribution (see Figure 3).

The estimated poses of the particles obtained in the proposal distribution can be updated if an image with an initialized object is available. If this is the case, the observation likelihood shown in Equation (3) should be evaluated by comparing actual and predicted observations. The actual observation in this research is provided by object segmentation using deep learning [49]. Estimating the predicted contour is explained in Section 3.3. This contour is built by finding the boundary around the projected object (onto the image). This projection is achieved using the predicted pose of the particle (obtained from the particle proposal) and the calibration matrix of the camera. It is important to note this process assumes that the object of interest is already initialized in the map.

In order to assess the observation likelihood, the distance between the actual and the predicted observations (often known as the residuals) should be measured. Defining this distance depends on the type of observations. For most classical solutions (such as ORB-SLAM), where the landmarks are points, this is simply measured as the Euclidean distance between the points. However, the contour-based method used in this research does not provide such a point-to-point correspondence. Thus, a novel method of measuring the distance using intersection over union (IoU) is developed. The IoU is measured using the observed contour ($z_{k}$) and the predicted contour (${\tilde{z}}_{k+1}$). The distance is inversely proportional to IoU, as shown in Equation (4). Finally, Equation (5) can be used, where the previous and current weights of the particles are denoted as $w_{k}^{\left[n\right]}$ and $w_{k+1}^{\left[n\right]}$. The exponent term on the right hand is maximum when the two contours exactly coincide, and it becomes smallest when one contour completely residues outside of the other. The symbol ղ denotes the normalizing term, and it ensures that the weight of the particles sums to one. More information about a fast-weighting process and estimating IoU is provided in Section 3.3. Figure 4 shows a schematic of the object and its segmented and predicted contours. The weight update is only possible if the object is already initialized in the map. For the object that is not initialized, a different process should be followed (this is explained in Section 3.4). It is important to note that in the case that no objects are detected in the image, no updates are available. Under these circumstances, the solution should rely on IMU mechanization, which can lead to the accumulation of errors in a very short time.
(4)$$d\left({\tilde{z}}_{k+1}^{\left[n\right]},z_{k+1}\right)=(Io{U({\tilde{z}}_{k+1}^{\left[n\right]},z_{k+1}))}^{-1}$$
(5)$$w_{k+1}^{\left[n\right]}=ղ\;w_{k}^{\left[n\right]}\left[exp\left(-\frac{1}{2\sigma_{d}}{\left(d\left({\tilde{z}}_{k+1}^{\left[n\right]},z_{k+1}\right)-1\right)}^{2}\right)\right]$$

### 3.3. Measuring IoU

As mentioned, the predicted projection of an object’s boundary should be estimated to weigh the particles. Unfortunately, projecting the 3D model of the object onto the image and then finding the boundary of the object is computationally costly since this process should be repeated for each particle. In the following passages, a method is introduced that will greatly decrease the runtime of the algorithm. As a test step, the algorithm projects the centroid of the 3D object onto the camera. As mentioned, this projection is achieved by using the associated predicted pose of the particle and camera calibration matrix. If the distance between this centroid and the centroid of the segmented object (observed contour) is larger, it is more likely that the particle has a higher error in the pose. Thus, such a particle can be assigned to a low weight without performing any additional step. In the second step, a downsampled 3D model of the object is projected onto the image. Similarly to the previous step, the centroid of these projected points is compared to the centroid of the segmented object. If the distance is larger than a threshold, a low weight to the corresponding particle is assigned.

The boundary around the projected points should be estimated for all the remaining particles. A possible approach is to use the alpha-shape [56]-based 2D boundary detection (e.g., as provided in MATLAB version 9.12 (R2022a)). This boundary can be rasterized (the boundary points can be inserted into an image, where the pixels belonging to the boundary are labelled as one, and the pixels belonging to the background are labelled as zero). It is possible that the rasterized boundary will be disconnected. In order to address this issue, efficient morphological operations such as dilation can be used to create a connected boundary. Further, the hole-fill operation [57] can be used to assign ones to the points inside the boundary. As the segmented object in the image is already in the binary format, the abovementioned steps can be skipped for this image. Finally, once the two masks are available (the observed and the predicted), the IoU can be calculated using efficient logical operations for binary images. The process explained above is provided in Figure 5.

### 3.4. Landmark Initialization

In this section, the landmark initialization is explained. The landmark initialization is required if the object of interest is observed for the first time in an image. The initialization in the object-level SLAM should estimate the object’s position and orientation in the map. In this research, the undelayed initialization technique is utilized, which can improve the accuracy of the solution by providing partial updates immediately after the object is observed for the first time. As a monocular camera cannot observe the distance to the object, the object initialization can only be achieved up to an unknown scale. In order to estimate the pose of the object using one image, only the contours of the objects are used. Therefore, this approach is shape-based.

The developed initialization method depends on object segmentation. Object segmentation can be obtained with the help of deep learning methods. In this research, a fully convolutional network (FCN) U-Net [58] is utilized for the segmentation. The data is synthesized for this network using the method developed in [49]. The precision of this approach is reported to be over 0.94 (and a recall of over 0.85) in many experiments.

The pose estimation is obtained in a coarse-to-fine process. The coarse pose estimation follows the approach in [49] and, for brevity, is not explained here. The output of the coarse process is a rough estimate of the pose of the object in the camera frame. This estimation is only accurate if the camera center, the object of the center, and the projected center of the object in the image are aligned.

In order to improve the pose estimated in [49], a refinement step is developed in this section. In the classical pose refinement, the features detected on the object are matched to the 3D model of the object, and these matched features are used to solve a Perspective-n-Point (PnP) problem. Unfortunately, since the developed method only assumes that the object’s contour is available, no features are detected on the object’s surface. In order to address this issue, the coarse pose estimation is used to project the object’s model (available in the CAD format) onto the image (see red object boundary in Figure 6). In the image, a feature correspondence is established between the two contours (the segmented and the projected). This correspondence is established by identifying points with the highest curvature. Equation (6) is used to estimate the curvature of the points on the contour. In this equation, c(x) and c(y) are the estimated boundary of the object. The symbols $x^{\prime }$, $x^{\prime \prime }$, $y^{\prime }$, and $y^{\prime \prime }$ correspond to the first and the second derivative with respect to the parameter of the curve (s) in the x and y directions.
(6)$$\kappa =\frac{\left|x^{\prime }(s)y^{\prime \prime }(s)-x^{\prime \prime }(s)y^{\prime }(s)\right|}{{\left({\left(x^{\prime }\left(s\right)\right)}^{2}+{\left(y^{\prime }\left(s\right)\right)}^{2}\right)}^{3/2}}$$

The identified high-curvature points on the boundary of the segmented object and the boundary of the projected object are utilized to establish a correspondence by first scaling and translating the projected boundary to the segmented object (see Figure 6, where the transformed and observed contours are shown in yellow and green). Once this transformation is achieved, the closest high-curvature points are declared as the corresponding features (the matched points are shown with numbers in Figure 6). The closest points can be found using nearest-neighbour methods such as kd-tree [59]. The established 2D-to-2D correspondence paves the path to establishing 2D-to-3D correspondence as well. This is possible as the correspondence between the project points and the 3D model is known. Finally, with the help of established 2D-to-3D correspondences, the pose of the object is refined using the P3P [60] and RANSAC [61]. The flowchart of the algorithm is provided in Figure 7.

### 3.5. Challenges Associated with Evaluation of Observation Likelihood

There are numerous challenges associated with the developed weighting process. These challenges are due to the utilization of the IoU for measuring the observation likelihood. The two most important challenges (Challenge I and Challenge II) are explained in detail in this section. Challenge I can occur when some of the pixels of the object are not identified during the object segmentation process. Possible reasons for this can be due to the occlusion of the object of interest, low-resolution images, or the long distance of the camera from the object of interest. Challenge I can frequently occur during the navigation. Figure 8 (first row) explains this problem schematically. In this figure, a schematic of the segmented object and the predicted object in the image is shown. The occluded area is shown with a rectangular grey box. It can be seen that once the object of interest is occluded, particles $p_{1}$ and $p_{2}$ will result in a similar IoU (and lower than 1). However, particle $p_{1}$ is much closer to the actual observation. Problems such as these will lead to assigning similar weights to better and worse particles, thus challenging the developed algorithm.

Challenge II can occur when pixels belonging to the background clutter or another object is detected as part of the foreground. This challenge can occur if similar-looking objects are close to the object of interest. However, in our experiments, Challenge II is much less likely to be encountered than Challenge I. In order to illustrate Challenge II schematically, Figure 8 (bottom row) can be inspected. In this figure, the nearby rectangular object is segmented mistakenly as a part of the object. Both particles $p_{1}$ and $p_{2}$ are assigned a large weight. However, particle $p_{1}$ is closer to the correct observation. Challenge II can lead to larger weights in comparison to Challenge I.

In order to identify if Challenge I or Challenge II has occurred, a fault detection algorithm can be used. As mentioned, most particles will be assigned to lower IoU values if Challenge I occurs. Simply by testing the particle with the maximum weight, it is possible to detect the occurrence of Challenge I. In such circumstances, the observation can be discarded in order to avoid introducing erroneous updates to the process of particle filtering.

The detected false positive pixels in the background due to Challenge II are often not close to the foreground (object). Assuming that this is the case, a simple binary pixel grouping method can be used. Further, as the largest group is most likely to correspond to the segmented object, the smaller groups can be removed. Thus, they cannot contribute to measuring the particles’ weight.

## 4. Results and Discussion

The experiments are performed using a designed handheld device. This device includes Xsens MTi-G-710 as the IMU sensor, which is a micro-electrical-mechanical system (MEMS) device. The noise specification of this sensor is reported as 60 μg/√HZ for the accelerometer and 0.01 degrees/s/√HZ for the gyroscope. The known six-position calibration method [51] can be used to estimate deterministic errors such as bias, scale and non-orthogonality. However, since the developed system is robust, non-Gaussian errors can also be mitigated in the process of filtering. Thus, for the purpose of the algorithm, identifying the bias of the accelerometer and gyroscope (which are the most significant deterministic errors) is sufficient. The defined handheld device also includes a monocular camera. The electrical board for this sensor is developed by Arducam, and the sensor itself is an 8 Mega Pixel IMX219 (which produces lower-quality images than most smartphones nowadays). The intrinsic parameters of the camera are obtained using MATLAB’s Camera Calibration toolbox. The extrinsic calibration parameters between the IMU and the monocular camera are derived from the CAD model of the system, and it is also tested using Kalibr [62]. Kalibr is a known algorithm to estimate the extrinsic calibration of an IMU and a monocular camera using special target boards. The data are collected on a Mini Desktop PC (BeeLink Mini S). Such Mini Desktop PCs are low-cost devices in comparison to smartphones. The specifications of the sensors and the calibration process are provided in Table 2. The image of the device is provided in Figure 9.

In the following, four sets of experiments are provided. In the first experiment, the coarse-to-fine pose estimation is assessed qualitatively. This process is independent of the RBPF-SLAM. In the second set of experiments, the performance of the RBPF-SLAM is evaluated. In the third experiment, the effect of increasing the number of particles on the accuracy of the solution is assessed. While all previous experiments were conducted using one object, the last experiment was conducted using two objects.

As mentioned, pose estimation is very important in the initialization of the object. In the first experiment, pose estimation is studied qualitatively. The camera is moved around the object of interest, and the pose is estimated in each image. The estimated coarse pose and the refined pose are used to project the 3D model of the object onto the image. The projected models are compared to the segmented object in the image. Results are provided in Figure 10. In this figure, the projected model using coarse estimation is shown in red. As discussed in Section 3.4, the coarse pose estimation always assumes that the camera center, the centroid of the 3D model of the object in the map, and the projected center of the object in the image all reside on a line. The refined pose is shown in purple in Figure 10 (the segmented object using deep learning is shown in yellow). This figure demonstrates that the coarse pose estimation is erroneous, while the final estimated pose is very close to the segmented object in the image. It is important to note that the refined pose is not a simple in-plane translation of the object in the image. This can be seen best in Figure 10 (third row/second column), where the object’s pose is refined substantially. Further, the accuracy of the initial pose will be improved in the process of the tightly coupled solution.

The performance of tightly coupled solutions is evaluated using different metrics. The accuracy of the method is estimated by measuring the distance between the initial pose and the final pose and compared to the ground truth value. Since the developed method outputs particles, a method should be used to estimate a single pose using all the particles. One possible approach is using the weighted pose of the particles. This is denoted as the expected value (EV) in this section and estimated using Equation (7). The second method of estimating the pose is selecting the particle with the largest weight. This is denoted as the maximum a posteriori (MAP) estimation. Further, the accuracy of pose estimation is reported in terms of absolute error (reported in cm) and relative error (reported as the ratio of the error to the total travelled path (TTP)).
(7)$$EV=\sum_{n\in \{1:N\}}w_{t}^{\left[n\right]}x_{t}^{\left[n\right]}$$

In addition to errors in the position, other parameters are also utilized to assess the performance of the developed method. These assessment metrics are the average IoU of the particles throughout the motion of the device. This value can be one at maximum and can assume a minimum of zero. The second assessment criterion is the failure rate. As mentioned in Section 3.5, the developed method can fail due to many challenges. In order to avoid these failures, a fault detection algorithm is used. The failure rate is defined as the number of epochs where the particle update failed (due to either Challenge I or Challenge II) to the total number of epochs. The last metric of assessment is the runtime of the developed method. The dataset is captured in two indoor environments with varied levels of background clutter. Sample images are provided in Figure 11. Further, the camera’s distance to the object is also varied in these experiments. Both the distance of the object and the background clutter can affect the accuracy of the object segmentation and thus affect the performance of the developed method.

The results are summarized in Table 3, Table 4 and Table 5 for five experiments. Based on these results, the accuracy of the developed method is in the range from 4.1 to 13.1 cm (0.005 to 0.021 of TTP) using EV. The error varies in the range of 18.9 to 35.7 cm (0.023 to 0.052 of TTP) using MAP. The accuracy of the solution is not affected by the background clutter, the distance of the object from the camera, and the length of the trajectory. Thus, the developed solution is not diverging with longer trajectories. Similar accuracy is expected to be achievable for even longer trajectories if frequent updates are available using the monocular camera. The IoU is in the range of 0.747 to 0.821 (this value can be considered high). During the navigation, a few failure rates are detected (0.00 to 0.01). It is important to note that even in failed cases, the solution did not diverge (see Test 5).

The estimated error ellipses and the robot’s trajectory are shown in Figure 12 and Figure 13 for Test 1 and 4, respectively. Figure 12a and Figure 13a show the trajectory and the camera’s positions in red and green, respectively. The initial and the final positions of the device are denoted as Start and End in this figure. In Figure 12b and Figure 13b, the particles are shown in green. Further, the error ellipsoids, including 50% of the total weight, are shown as well. The figures show the uncertainty in the direction from the camera center toward the object is the highest.

The developed method can be compared to a similar IMU/camera fusion method in [63], where the reported error ranges from 0.01 to 0.037 (error ratio to TTP). This indicates a similar performance to the developed solution in this research. It is important to note that the method in [63] utilizes a stereo camera rig, while our solution utilizes a monocular camera.

In order to study the effect of the number of particles, experiments are performed by modifying the number of particles. The experiment’s results are summarized in Table 6. Based on these experiments, changing the number of particles from 5000 to 19,000 did not seem to change the accuracy significantly. However, the runtime of the algorithm has increased substantially (from 6149.2 s to 13,390.3 s).

The objects can be severely occluded during the motion of the device. This will affect the performance of the solution. In the example shown in Figure 14, the object of interest (the green model of Stanford’s Bunny) is occluded by another object in some viewpoints. The estimated particles are shown in the plot on the right-hand side during the corresponding epochs inside a red box. It can be seen that based on this plot, when the object of interest is occluded, the uncertainty in the solution has increased (as the error ellipses become more elongated). However, the algorithm does not diverge, and the uncertainty decreases when the object is not occluded in the later epochs.

The abovementioned tests have only utilized a single object to provide an object-level solution to the SLAM problem. While these results can provide evidence that the developed algorithm can produce relatively high accuracy in many circumstances, relying on a single object might lead to errors in real-life scenarios. For example, the object of interest may become occluded longer during the navigation. Therefore, relying on more than a single object can help improve the robustness of the algorithm to the errors caused by Challenge I and Challenge II. In the following passages, the results for using two objects are provided. The main difference between this and the previous experiment is that more than a single landmark initialization should be performed (one for each new object)

Throughout the navigation, there are two different cases that can occur when multiple objects are used. The first case occurs if only a single object is observed in the image. In this case, a data association procedure should be used to determine which landmark in the map the observation is associated with. Once such data association is performed, the following steps (particle proposal and the weight updates) are the same as using only a single object. In the second case, more than one object is observed in the images. In such circumstances, the IoU for the individual object should be estimated, and the object with the highest IoU should be used to update the particles’ weight. This approach is not the only possible weight update strategy. However, since the developed solution can frequently encounter Challenge I (e.g., when one or more objects are occluded), using a single object with the highest IoU can improve the robustness of the developed solution. The results for the two objects are provided in Table 7. Based on these results, it can be seen that the error and IoU did not change using multiple objects in comparison to using only a single object.

The developed method has been tested in many experiments. The illumination conditions in these experiments are different due to the different number and types of light sources. Further, the experiments are conducted at different times of the day (which can influence the level of ambient light). However, the influence of illumination conditions on the accuracy of the solution is not directly investigated. In order to achieve this, other variables (e.g., camera distance to the object, background clutter) should be kept unchanged for such experiments.

The pose refinement step relies on finding the highest curvature points along the object’s contour. However, reliable detection of such points for some objects is not possible (for example, a ball-shaped object). In these circumstances, the proposed pose refinement should be avoided, and the coarse pose estimation should be directly used for the initialization. A comparison between the obtained accuracy with and without pose refinement should be conducted in future studies.

The experiments above assume that landmarks in the map (here, objects) are static. In indoor environments, this assumption is valid for most objects. However, dynamic objects should be detected and excluded from the process of the proposed solution if they are present in the environment.

## 5. Conclusions and Future Work

In recent years, object-level solutions to the SLAM problem have emerged. An object-level solution paves the path for the interaction of a robot with the objects in an environment. In the past, many object-level solutions depended on Gaussian error assumption. Furthermore, these solutions often relied on ORB-SLAM 2 and do not offer a standalone object-level solution. The developed approach in this research shows that a standalone object-level solution to the SLAM can be provided using RBPF-SLAM. The important advantage of this approach is that no assumption about the distribution of the errors is made.

Object representation is an important component of an object-level SLAM. The state-of-the-art methods rely on simple geometrical shapes (e.g., ellipsoids, cuboids) to represent an object. However, these simpler forms are not suitable for many applications. Another commonly used representation method is based on feature points. Detection of such feature points requires textures on an object’s surface. In this research, instead of relying on simpler shapes or salient feature points, the objects are represented using 2D shape priors. The reliance on 2D shape priors is very useful when object segmentation is used. This is due to the fact that segmentation methods can only identify the contour around the object. With a shape-based object representation, we have developed a novel particle weighting method using IoU that does not require point-to-point feature correspondence.

In the past, the fusion of an IMU and a monocular camera has been achieved using many methods. In these solutions, the IMU is used to predict the trajectory of a device, while the camera is used to correct the prediction and build a map of the environment. However, most of the state-of-the-art approaches rely on delayed landmark initialization. During the delayed period, the solution can suffer from error accumulation. In contrast to these methods, the solution offered in this research is based on undelayed initialization. This approach can immediately provide updates to the position of the device once the objects are observed for the first time. Thus, it avoids relying only on the IMU mechanization for longer periods.

Based on the results shown, an error of 4.1 to 13.1 cm (0.005 to 0.021 of the TTP) can be achieved with the designed handheld device. It is also demonstrated that the IoU of the developed method is about 0.8. The failure rate of the algorithm is very low (within 0.00 to 0.03). The solution seems reliable under many different circumstances (such as the presence of an occlusion). Further, the solution is not degraded as the trajectory becomes longer. Similar results are expected to be obtained regardless of the trajectory length if an object is visible in the images. Finally, the effect of changing the number of particles is also investigated, and it is shown that increasing the number of particles to more than 5000 does not decrease the errors in the estimated trajectories

In future research, it is important to reduce the computation cost further. One possible solution is a fusion with a rangefinder, such as an ultrasonic sensor. This fusion can help discard the particles that do not fall within a threshold of the observed distance from the object of interest. This elimination will help avoid calculating observation likelihood for many particles with inaccurate poses and thus substantially contribute to the computational cost reduction.

## Figures and Tables

**Figure 1 sensors-23-07958-f001:**
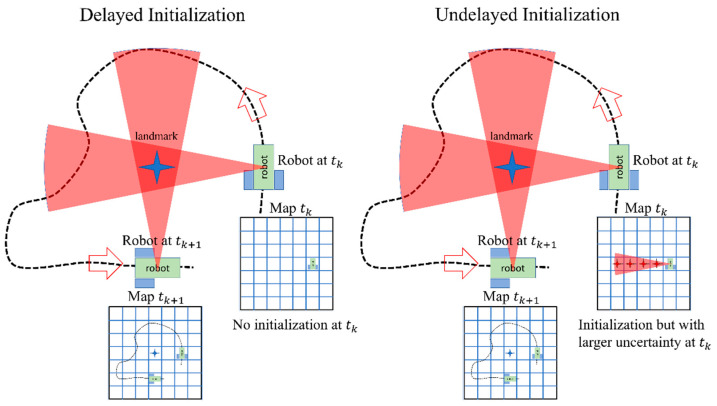
A comparison of delayed and undelayed initialization.

**Figure 2 sensors-23-07958-f002:**
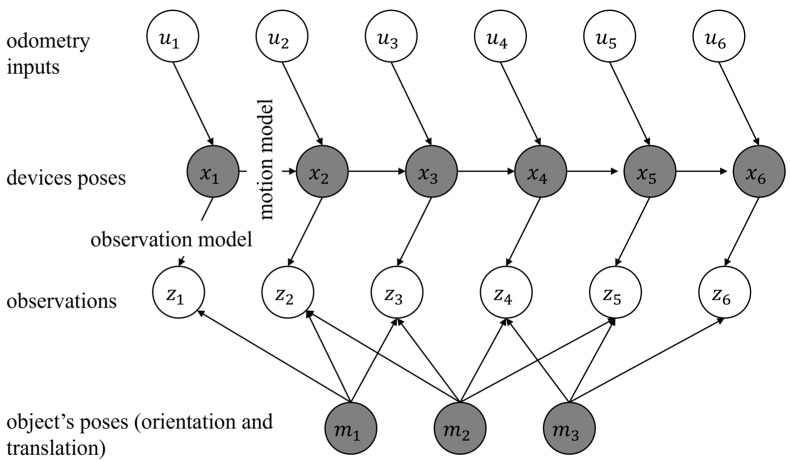
The formalization of the SLAM problem as DBN.

**Figure 3 sensors-23-07958-f003:**
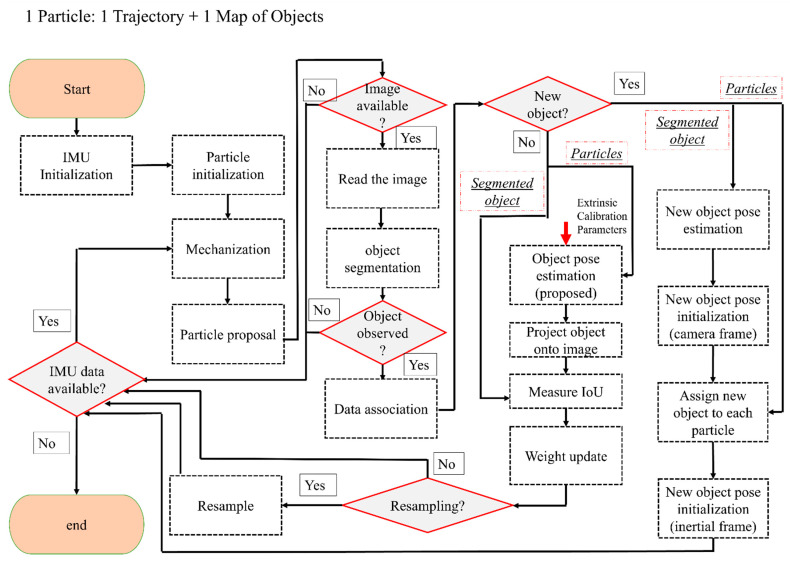
The proposed IMU/monocular tightly coupled solution with initialization to SLAM.

**Figure 4 sensors-23-07958-f004:**
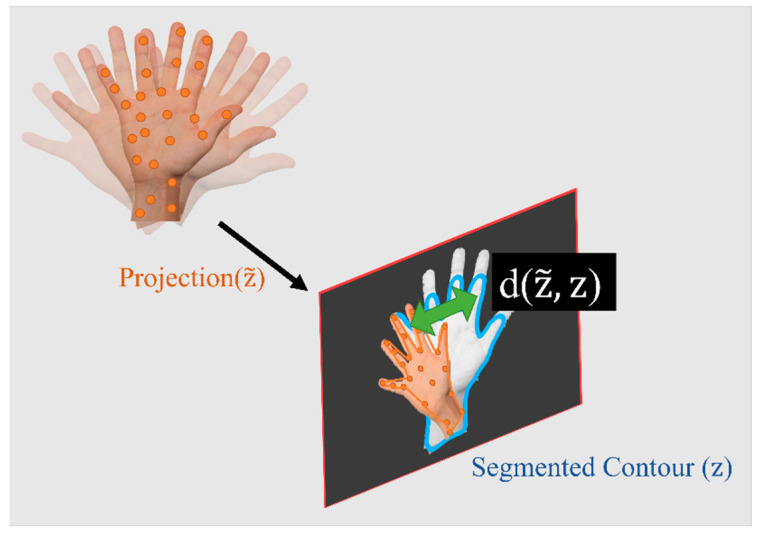
An illustration of the defined distance between predicted and projected contours.

**Figure 5 sensors-23-07958-f005:**
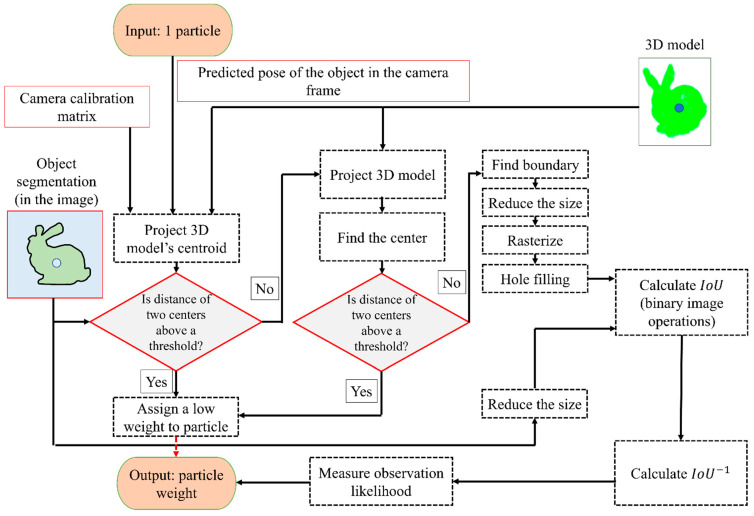
The flowchart of the fast particle weighting process.

**Figure 6 sensors-23-07958-f006:**
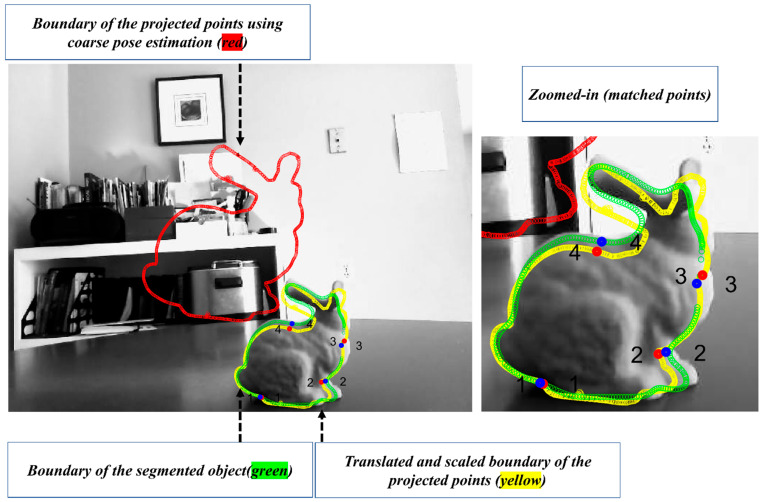
Illustration of the procedure to establish point correspondences for pose refinement.

**Figure 7 sensors-23-07958-f007:**
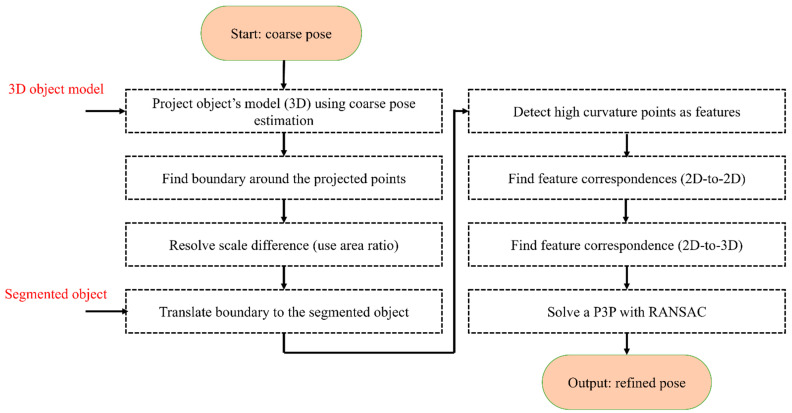
The flowchart of coarse-to-fine pose estimation.

**Figure 8 sensors-23-07958-f008:**
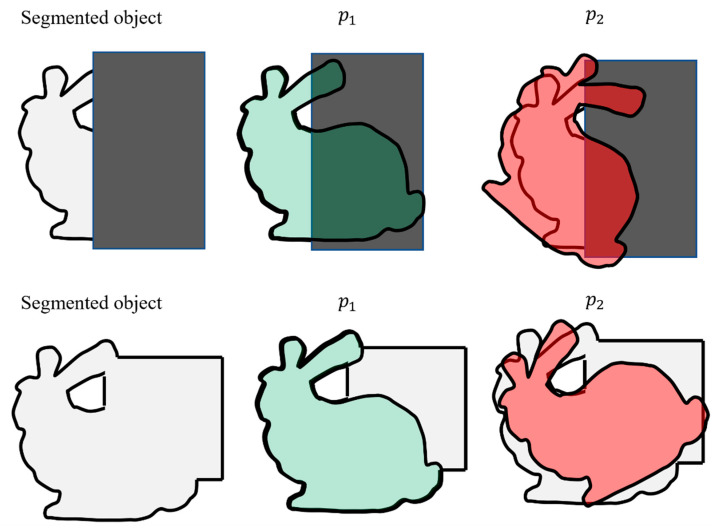
Schematics of the challenges with the observation likelihood. The **top row** shows Challenge I (the dark grey box represents an occlusion), and the **bottom row** shows Challenge II.

**Figure 9 sensors-23-07958-f009:**
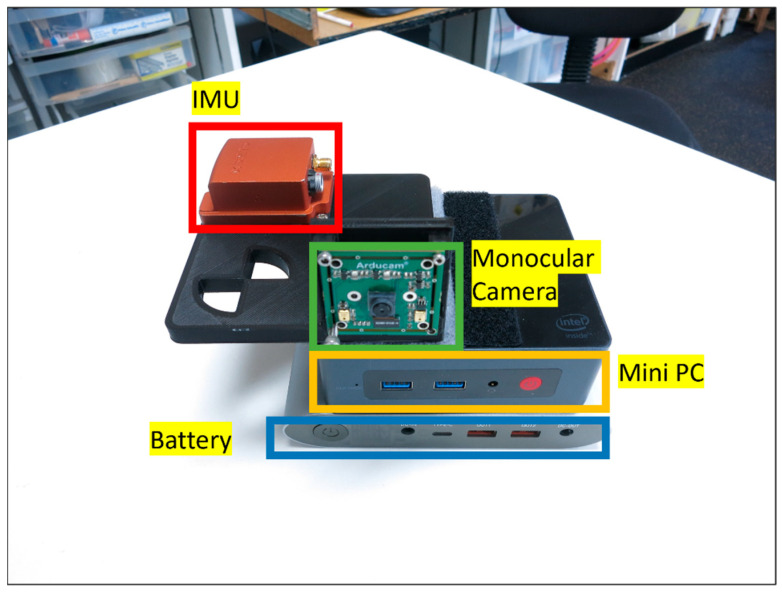
Illustration of the handheld device used for the experiment.

**Figure 10 sensors-23-07958-f010:**
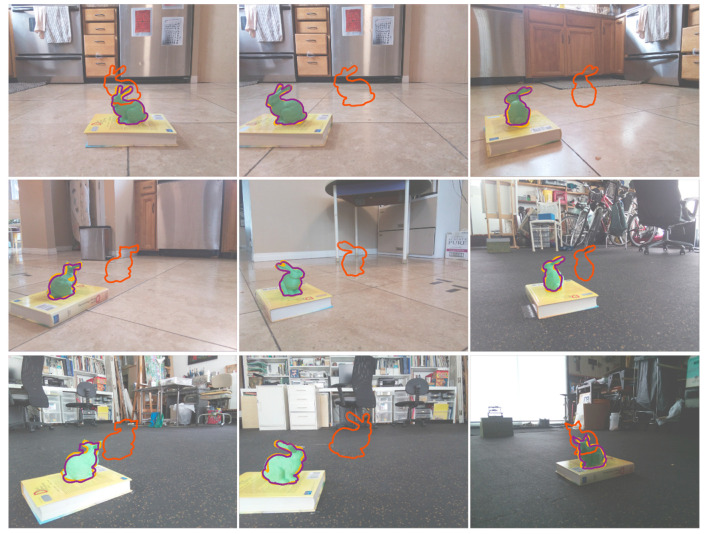
Qualitative comparison of the projected object before and after pose refinement.

**Figure 11 sensors-23-07958-f011:**
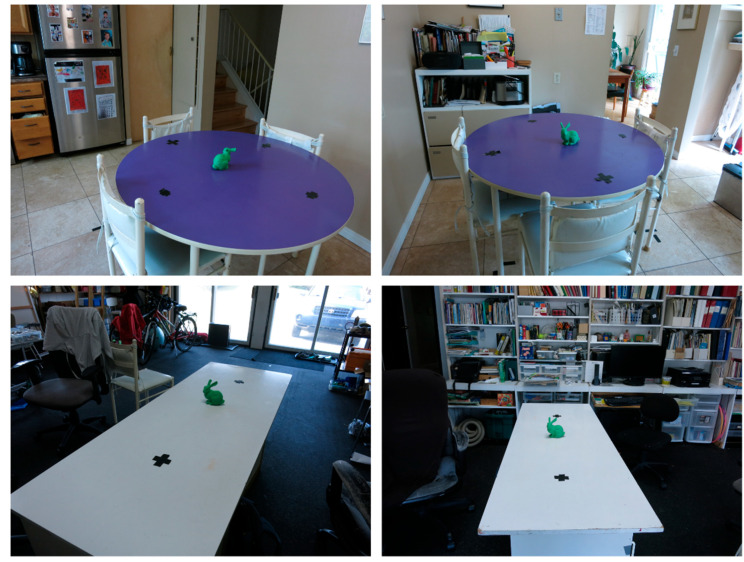
This figure illustrates sample images of the environments of the experiments. The marked points indicate the initial and final positions of the device.

**Figure 12 sensors-23-07958-f012:**
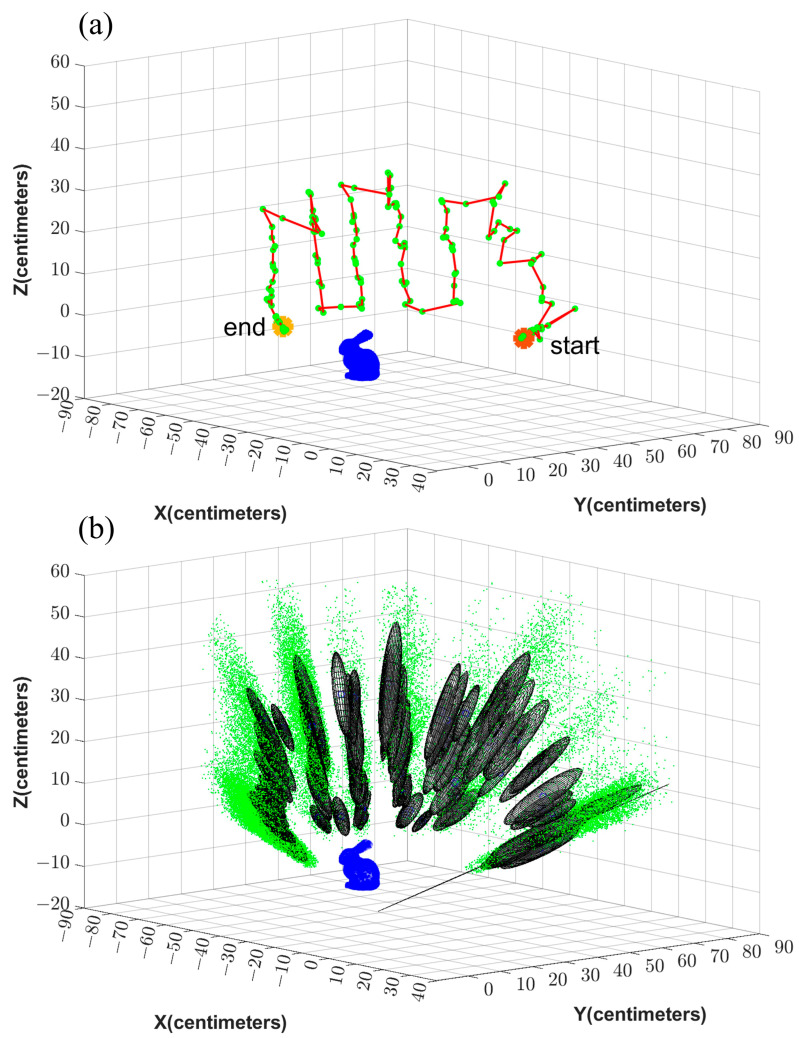
(**a**) illustrates the trajectory (red) and the locations where the images are captured (green). (**b**) illustrates the corresponding particles (green) and the error ellipses.

**Figure 13 sensors-23-07958-f013:**
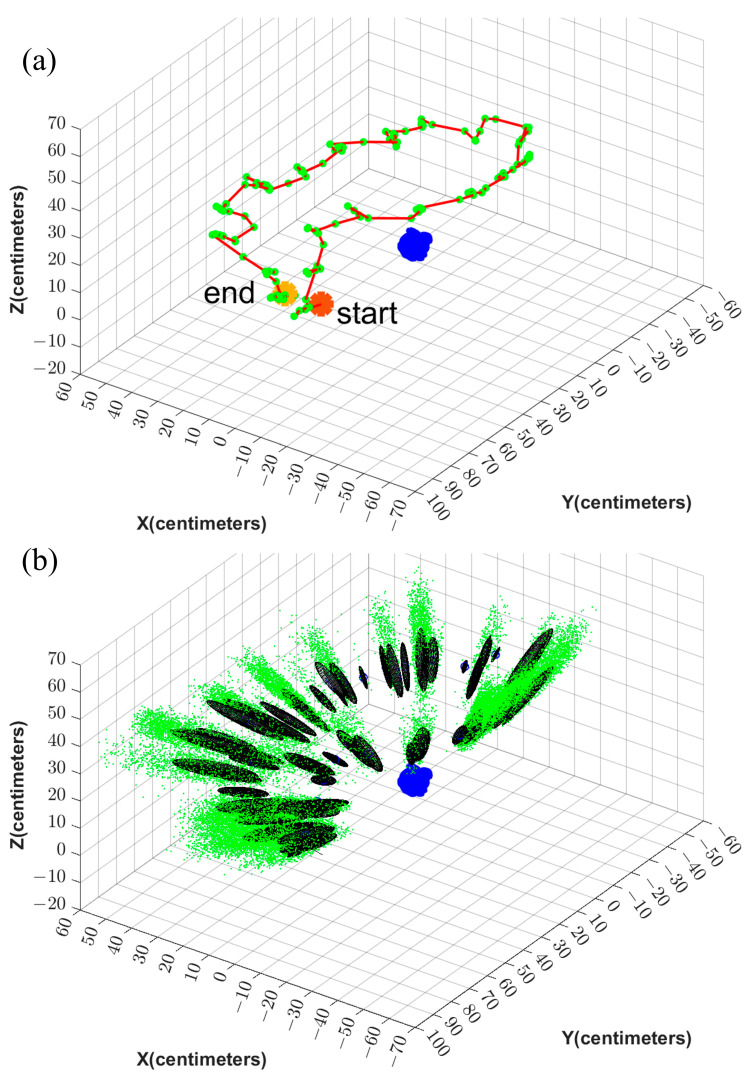
(**a**) illustrates the trajectory (red) and the locations where the images are captured (green). (**b**) illustrates the corresponding particles (green) and the error ellipses.

**Figure 14 sensors-23-07958-f014:**
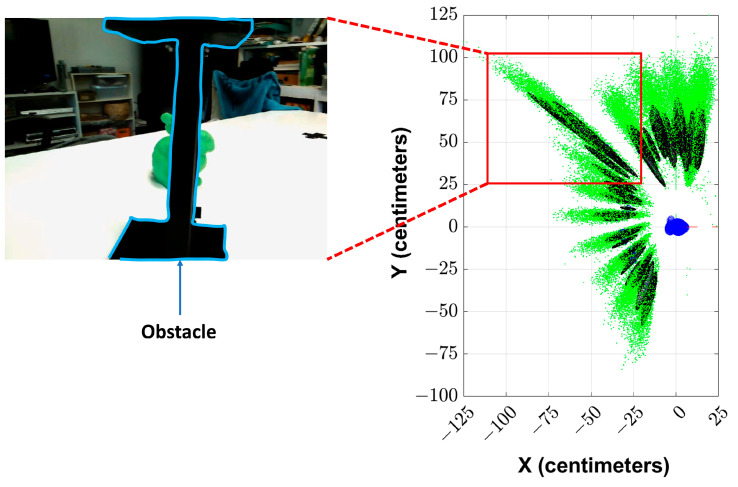
The figure illustrates an increase in uncertainty when the object is severely occluded. The obstacle’s boundary is highlighted (blue) in the image shown on the left.

**Table 1 sensors-23-07958-t001:** Summary of some of the selected object-level SLAM methods.

Reference (Date)	Object Detection	Object Representation	Frontend/Backend	Relies on ORB SLAM 2 (or 1)?
[22] (2011)	Detector/Descriptor	Offline (feature points)	EKF/no backend	No
[10] (2015)	Detector/Descriptor	Online (feature points)	LBA/FG	Yes
[12] (2016)	Detector/Descriptor	Offline (feature points)	LBA/graph optimization	No
[27] (2017)	Detector/Descriptor + DNN	No 6DoF object models used	LBA/FG	No
[44] (2018)	DNN	Offline (3D shape priors)	Decoupled estimation/FG	No
[28] (2018)	DNN	Ellipsoids	Decoupled estimation/FG	No
[29] (2019)	DNN	Ellipsoids	LBA/GBA	No
[31] (2019)	Detector/Descriptor + DNN	Cuboids	LBA/GBA	Yes
[30] (2020)	Detector/Descriptor + DNN	Ellipsoids	LBA/FG	Yes
[47] (2021)	DNN	Offline (3D shape priors)	Decoupled estimation/FG	Yes
[19] (2021)	DNN	3D models with RGB values	RBPF	No
[14] (2023)	DNN	Online (3D shape priors)	LBA/GBA	Yes
Ours	DNN	Offline (2D shape priors)	RBPF	No

**Table 2 sensors-23-07958-t002:** A summary of the sensors, processor, and power supply used for the experiments.

Sensor	Intrinsic Calibration	Extrinsic Calibration	Additional Information
Monocular camera	MATLAB’s calibration toolbox	Provided by 3D CAD model	Arducam with 8MP IMX219
IMU	Six-position static calibration	Provided by 3D CAD model	Xsens MTi-G-710
Power supply	N/A	N/A	Krisdonia Laptop Power Bank
Mini Desktop PC	N/A	N/A	Beelink Mini S

**Table 3 sensors-23-07958-t003:** The errors in the position (cm) for five tests.

Test	Error in Position		Distance (Camera to Object)	Experiments Details
	EV (cm)	MAP (cm)		
Test 1	7.8	22.4	short	lower clutter, shorter trajectory
Test 2	4.1	18.9	long	lower clutter, shorter trajectory
Test 3	12.3	35.7	medium	higher clutter, longer trajectory
Test 4	13.1	32.7	medium	higher clutter, longer trajectory
Test 5	12.3	25.5	medium	higher clutter, average trajectory

**Table 4 sensors-23-07958-t004:** The ratio of the error in the position to the TTP.

Test	Error in Position		Distance (Camera to Object)	Experiments Details
	EV/TTP	MAP/TTP		
Test 1	0.015	0.043	short	lower clutter, shorter trajectory
Test 2	0.005	0.023	long	lower clutter, shorter trajectory
Test 3	0.018	0.052	medium	higher clutter, longer trajectory
Test 4	0.021	0.052	medium	higher clutter, longer trajectory
Test 5	0.014	0.029	medium	higher clutter, average trajectory

**Table 5 sensors-23-07958-t005:** The IoU of the particles along the trajectory.

Test	Other Assessment Metrics		Distance (Camera to Object)	Experiments Details
	IoU	Fail Rate		
Test 1	0.805	0.00	short	lower clutter, shorter trajectory
Test 2	0.805	0.00	long	lower clutter, shorter trajectory
Test 3	0.819	0.00	medium	higher clutter, longer trajectory
Test 4	0.821	0.00	medium	higher clutter, longer trajectory
Test 5	0.747	0.01	medium	higher clutter, average trajectory

**Table 6 sensors-23-07958-t006:** The performance of the developed tightly coupled solution for different particle numbers.

Number of Particles	Error in Position		Other Assessment Metrics		
	EV (cm)	EV/TTP	IoU	Time (s)	Fail Rate
5000	12.4	0.018	0.813	6149.2	0.00
7000	11.4	0.020	0.821	8577.4	0.00
9000	10.9	0.019	0.824	9291.8	0.00
13,000	10.8	0.018	0.830	11015.2	0.00
17,000	13.4	0.022	0.835	12488.7	0.00
19,000	11.0	0.018	0.837	13390.3	0.00

**Table 7 sensors-23-07958-t007:** The performance of the developed tightly coupled solution using two objects.

Test	Error in Position				Other Details	
	EV (cm)	MAP (cm)	EV/TTP	MAP/TTP	IoU	Fail Rate
Test 6	9.1	17.3	0.012	0.023	0.815	0.03

## Data Availability

The data presented in this study are available on request from the corresponding author.

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
