# Peer review of "Shaped-Based Tightly Coupled IMU/Camera Object-Level SLAM"

_sensors, 2023, doi:10.3390/s23187958_

Round 1

Reviewer 1 Report

An accurate object-level solution to the SLAM problem with a 4.1 to 13.1 cm error in the position (0.005 to 0.021 of the total path) has been presented in this work based on Raoblackwellized Particle Filtering (RBPF) that does not assume predefined error distribution for the parameters. The solution relies on the shape and thus can be used for objects that lack texture on their surface and the developed tightly coupled IMU/camera solution is based on an undelayed initialization of the objects in the map. With the work above in this paper, the contributions are sufficient to make this work publishable in the journal. It is well-written and organized. I have the following suggestions to improve the paper:

(1) In the introduction, please highlight the contributions of this work instead of only stating what has been done in this paper in the last paragraph of Section.1.2.

(2) In the introduction, you have described the issue of measurement delay and the accumulation error of dead reckoning. This has also been a persistent challenge in improving positioning accuracy based on IMU/GPS fusion. Fortunately, an alternative approach has been employed in: automated vehicle sideslip angle estimation considering signal measurement characteristic. In this study, the issue of GNSS signal delay has been effectively tackled through an estimation-prediction integrated framework. Additionally, the integration of reverse smoothing and grey prediction is utilized to mitigate cumulative velocity errors during relatively low sampling intervals of GNSS data. As a result, this framework holds significant relevance for the IMU/camera integrated system and should be included and discussed accordingly.

(3) What will happen if the object is dynamically moving? Will the movement affect the SLAM performance?

(4) Regarding feature extraction, the camera is a good sensor but it lacks depth information. With the development of sensing technology, other sensors such as LiDAR and radar can also be used to localize the vehicle. Useful features can be leveraged to enhance the SLAM performance. For instance, in the work: integrated inertial-LiDAR-based map matching localization for varying environments; authors use expert knowledge to extract the feature in each LiDAR frame, then, a multi-sensor fusion scheme is used to achieve the localization. Please discuss this work in the paper in particular and mention it in the potential future work of this paper.

(5) Last minor comment, please adjust the font size of the text in the figures to match the main text in the paper. 

Reviewer 2 Report

1. Literature review section must also be extended. A comparative study may also be shown in graphical form.

2. Relevant references should be cited in the text body and all references should be in properly/sequentially arranged as cited in the text. Add/cite recent publication (2019, 2020, 2021) preferably.

3. The introduction includes a lot of work that is not necessary. Please include the problem statement, aim, objective, approach, key contributions, novelty and structure of the article.

4. For citations in the text, use square brackets and consecutive num­bers would write [1-5] but [1], [3], [5], etc.

5. The Limitations of the proposed study need to be discussed before conclusion.

6. Are the conclusions consistent with the evidence and arguments presented and do they address the main question posed?

7. What specific improvements should the authors consider regarding the methodology? What further controls should be considered?

8. What does the research address the main question? Do you consider the topic original or relevant in the field? Does it address a specific gap in the field?

1. Language must be improved as there are linguistic errors at some places.

Reviewer 3 Report

The manuscript proposed shaped-based tightly coupled IMU/Camera Object-level SLAM.  The developed solution is based on Raoblackwellized Particle Filtering (RBPF) and object shape information. The overall presentation is clear and the indoor localization results are good. Several issues:

1. here a rabbit object is used, its contour is complex and easy to detect key points,   suppose a basketball or football is here, your current method is still ok or not?  if not, are there some possible methods to solve this?

2.  not sure if lighting will influence or not?

3.  Conclusion is too long, some content can be put in discussion. you could add a discussion suggestion.  illustrated the restrictions of your methods.

English level is good

Round 2

Reviewer 1 Report

Thanks for the reply.

Reviewer 2 Report

After reviewing all the suggestions made, I observed that they were fully carried out. Therefore, I have no more suggested changes.